# Direct Chill Casting and Extrusion of AA6111 Aluminum Alloy Formulated from Taint Tabor Scrap

**DOI:** 10.3390/ma13245740

**Published:** 2020-12-16

**Authors:** Kawther Al-Helal, Jayesh B. Patel, Geoff M. Scamans, Zhongyun Fan

**Affiliations:** 1Brunel Centre for Advanced Solidification Technology, Brunel University London, London UB8 3PH, UK; jayesh.patel@brunel.ac.uk (J.B.P.); zhongyun.fan@brunel.ac.uk (Z.F.); 2Innoval Technology Limited, Beaumont Cl, Banbury OX16 1TQ, UK; geoff.scamans@innovaltec.com

**Keywords:** taint tabor scrap, recycling, high shear melt conditioning, direct chill casting, extrusion, PCG, paint bake simulation, cold rolling

## Abstract

AA6111 aluminum automotive body-sheet alloy has been formulated from 100% Taint Tabor scrap aluminum. Direct chill casting with and without high shear melt conditioning (HSMC) was used to produce the AA6111 alloy billets. Both homogenized and non-homogenized billets were extruded into sheets. The optical micrographs of the melt conditioned direct chill (MC-DC) samples showed refined equiaxed grains in comparison to direct chill (DC) cast and direct chill grain refined (DC-GR) samples. Optical metallography showed extensive peripheral coarse grain (PCG) for the DC, DC-GR and MC-DC planks extruded from the homogenized standard AA6111 billets while planks extruded from modified AA6111 billets (with recrystallization inhibitors) showed thin PCG band. The co-addition of recrystallization inhibitors Mn, Zr, and Cr with elimination of the billet homogenization step had a favorable impact on the microstructure of the AA6111 alloy following the extrusion process where a fibrous grain structure was retained across the whole section of the planks. The mechanical properties of as-cast planks extruded from non-homogenized billets were similar to those extruded from homogenized billets. Eliminating the homogenization heat treatment step prior to extrusion has important ramifications in terms of processing cost reduction.

## 1. Introduction

There is a growing demand to recycle aluminum end-of-life scrap to save natural resources, reduce production energy, and cut CO_2_ emission. The metal extraction for primary aluminum production consumes about 3.5% of global electricity and emits 1% of the total CO_2_ emission. The challenge for aluminum production is to cut 50% of CO_2_ emission by 2050. Re-melting of end-of-life aluminum scrap consumes 5% of the energy required to produce primary aluminum and can saves up to 95% of CO_2_ emission [1,2]. In the UK, the main motivation for recycling is the provision of aluminum automotive body-sheet and structural castings from a sustainable end-of-life scrap source for the mass manufacture of lightweight low embedded carbon vehicle bodies.

In the automotive industry, AA6111 is an automotive body-sheet aluminum alloy that used for structural applications [3]. It was developed by Alcan’s research laboratories in Canada. The AA6111 alloy is used in aluminum intensive models developed by Jaguar Land Rover, with copper as an alloying addition (copper level of 0.5 to 0.9 wt.%) and is significantly stronger than AA6016. In addition to the chemical composition, the thermo-mechanical treatment such as annealing, work and a stabilizing precipitation-hardening step can provide the alloy with specific properties such as formability and bendability that can be developed in the high strength condition after the paint bake treatment or by deliberate final aging, higher strength, formability and hemming quality [4]. The heat treatable AA6111 aluminum alloy has an excellent combination of formability and paint bake response with good surface quality after the stamping and paint bake processes [5,6]. The stamped and assembled automotive body panels are finished through the paint bake heat treatment cycle used to cure both paint and the adhesives used for structural bonding. For good formability, AA6111 are supplied in the low strength T4P temper to ensure a low yield strength for successful stamping. The yield strength or dent resistance of the stamped panels is then increased by precipitation-hardening that occurs during the paint bake process [7,8].

In this study, AA6111 was formulated from 100% Taint Tabor scrap with no adding of primary aluminum. In the UK, Taint Tabor (TT) scrap consists of two or more aluminum alloys and it is free of wire or any non-metallic contamination [9]. Further separation and pre-treatment is then needed in processing the sorted aluminum scrap and before casting into Recycled Secondary Ingots (RSIs). Various sorting operations can be used in separation of wrought aluminum scrap from other non-ferrous metals and plastics such as floatation, magnetic field, eddy current, or automatic sortation using advanced sensing and sorting technologies. The TT supplied for this work by Axion Group UK was from the standard feedstock of old rolled aluminum that is mainly construction scrap. Prior to ingot casting at Norton aluminum, the TT was shredded and processed by hand sorting, magnetic and eddy current separation by Axion [1].

Because of the disadvantage of using grain refinement in casting some of aluminum alloys, other techniques have been tested such as electromagnetic mixing, ultrasonic and the melt conditioning [10]. At Brunel Centre for Advanced Solidification Technology (BCAST), intensive melt conditioning technology was developed and combined with direct chill casting and twin roll casting for microstructural improvement [11]. The HSMC technology is a multi-purpose technology, which uses a rotor-stator mixing device and has been tested for grain refinement, metal-matrix composites casting, melt degassing and de-ironing [12]. Large-scale billets have been produced by applying intensive melt conditioning in the direct chill casting process. During melt conditioning direct chill process (MC-DC) process, the high shear device sucks the liquid metal and subject it to intensive shearing in the gap between the rotor and stator. The fine jet of sheared melt forced through the holes of the stator at very high velocity. The fine jet of liquid melt will form a unique macroscopic flow pattern in the sump, as shown in Figure 1. The sucking action of the rotor-stator device induces an intensive macroscopic flow in the lower part of the sump more than that in the upper part. The combination of both dispersive and distributive mixing actions is the major advantage of applying the high shear device while keeping the melt surface relatively stable. The surface stability of melt in the sump will help to avoid any possibility of oxides entrainment, where these oxides increase the hydrogen intake from the atmosphere and increase porosity [13,14,15,16].

Chemical segregation, hot tearing, porosity and non-uniform microstructure are the main defects in direct chill casting [17]. In this work, AA6111 billets were produced by using both the MC-DC casting technology and by conventional grain refinement (DC-GR) followed by extrusion and rolling.

The aim of this study was to assess the microstructure and tensile properties of AA6111 automotive body-sheet formulated from TT scrap. The end-of-life (TT) scrap was processed into billets in direct chill casting followed by extrusion process, cold rolling, heat treatment, and paint bake simulation.

## 2. Experimental Procedures

### 2.1. Billets Casting and Extrusion

For direct chill casting with and without high shear melt conditioning (HSMC), three batches of TT scrap (A052, F255 and F256) were received as RSIs from Norton aluminum (Cannok, UK). To match the target AA6111 composition, the first batch of TT (A052) was melted and formulated by adjusting Cu and Mg level only. The other two batches of TT (F255 and F256) were melted and alloyed with Cu, Mg, Mn, Cr, and Zr. The aim of adding recrystallization inhibitors Mn, Cr, and Zr to TT-6111 was to minimize the development of a coarse outer band of recrystallized grains during extrusion. Table 1 lists the chemical composition of the as-received Taint Tabor batches after re-melting at Norton aluminum, the formulated TT-6111 (1, 2 and 3) alloys and the target AA6111 compositions.

The billets casting of TT-6111 alloys were carried out in a conventional direct chill caster assembled with a tilting furnace and launder to pour the melt in the open-hot-top of the system. The casting speed of 152 mm in diameter billets was 110 mm/min and the water flow rate was 116 L/min. The melt temperature in the launder was about 690 °C. ‘Optifine’ (MQP, Whitwick, UK) (Al–3Ti–1B) grain refiner rod was used for the grain refined direct chill (DC-GR) casting. It was added to the melt in the launder before pouring into the sump. For casting with high shear melt conditioning (as shown in Figure 1), the rotor-stator device was preheated electrically to 700 °C and mounted on top of the hot-top. As soon as reaching 110 mm/min steady casting speed, the high shear rotor-stator device was immersed into the sump and switched on to run at speed around 2000 rpm. The billet length for each casting was ~2 m. The billets of TT-6111 (1) and modified TT-6111 (2) alloys were homogenized at 530 °C for 4 h followed by air cooling. For the processing of TT-6111 (3) billets, the homogenization step was eliminated, and the billets were extruded directly. All billets of the three batches were extruded into 118 mm width and 4.8 mm in thickness planks at a 530 °C and the speed was 3 m/min. A 16 MN long stroke direct extrusion press was used for extrusion of 152 mm diameter and 450 mm length billets. The direct chill casting (DC, DC-GR and MC-DC), billets homogenization and extrusion process were carried out at the Advanced Metal Casting Centre (AMCC) at Brunel University London. The billets and the extruded planks are shown in Figure 2.

### 2.2. Microstructure Characterization

Metallographic analysis was carried out on samples cut from different positions across the billets cross-section after casting and in the longitudinal section for extrusion flat bars. These samples were mounted in Bakelite resin at 180 °C, prepared by following the standard technique of grinding with SiC abrasive papers and polishing with 0.04 µm suspension of silica in water. The polished samples were anodized in Barker’s reagent at 20 V for 55 s to reveal the grain morphology and grain size for each casting condition. A Zeiss Axio-Vision microscope was used for analyzing the microstructure for different cross-sections and casting conditions of the samples. The average grain size was measured by using linear intercept method on polarized images.

### 2.3. Thermo-Mechanical Treatment and Tensile Testing

To measure the tensile properties after forming and paint bake treatment and/or after aging to peak properties, the extruded 4.8 mm thick 118 mm wide planks of TT-6111 (2) and TT-6111 (3) for all direct chill casting were selected from the center position of extrusion cast. The TT-6111 (1) planks were not considered suitable for further processing due to excessive peripheral coarse grain. The TT-6111 (2) and TT-6111 (3) planks were annealed for 2 h at 360 °C followed by cold rolling down to 3.0 mm. One half of each 3.0 mm planks were annealed at 360 °C for 2 h followed by cold rolling down to 1.7 mm thickness. One half of each 1.7 mm plank was then cold rolled down to 1.0 mm. Sheet cut from the rolled planks at each gauge was used for the machined tensile specimens. Simulation of industrial heat treatment of AA6111/T4P temper strip was carried at each gauge by the following steps: (**a**) solution heat treatment at 560 °C for 10 min; (**b**) cold water quenching and (**c**) immediate pre-age at 85 °C for 5 hr to develop the T4P temper.

Tensile specimens of as extruded and T4P temper planks of the three different gauges, casting conditions, and orientations were machined with 10 specimens for each condition. For development of the T8 temper, a 2% stretch and a paint bake simulation at 180 °C for 30 min was carried on half of the T4P temper tensile specimens. The mechanical properties of as extruded planks were measured on samples cut 0°, 90° and 45° orientations while for T4P and T8 temper in transverse and longitudinal orientation. Figure 3 shows the dimensions of tensile sample according to BS EN ISO 6892-1 standard. An Instron 5569 (Norwood, MA, USA) universal electromechanical testing machine was used for the tensile tests. This machine is equipped with Bluehill software v. 1.8.289 and with ±50 kN load cell. Tensile tests were conducted at room temperature with strain rate set to 1 s^−1^ until failure.

## 3. Results and Discussion

### 3.1. Microstructure Analysis

#### Direct Chill Casting Billets

In comparison to DC and DC-GR, the optical metallography of MC-DC castings for the three batches of TT-6111 (1), TT-6111 (2) and TT-6111 (3), showed the significant effect of high shear melt conditioning in grain refining with homogenous grain size distribution across the billets. The cast of modified TT-6111 (3) provided the best extruded plank microstructure and the peripheral coarse grain was eliminated. Figure 4 shows the microstructure of DC, DC-GR and MC-DC castings of modified TT-6111 (3) alloy of batch 3 (F256). The optical metallography of DC-GR and MC-DC cast billets showed refined equiaxed grains with defect free for the modified TT-6111 alloy in comparison to DC casting. The DC casting showed equiaxed large grains at the center of billet with columnar grains for the rest of the billet cross-section. Applying intensive melt shearing improved the grain morphology, reduced the grain size and enhanced the grain size distribution across the cast billets. For DC-GR and MC-DC billets, the average grain size was 220 ± 6.5 µm and 205 ± 3.1 µm, respectively. The equiaxed morphology with homogeneous grain size distribution are important for the quality of final product after extrusion and thermo-mechanical treatment.

Processing the recycled aluminum alloys with intensive melt shearing in direct chill casting results in high grain refinement and a homogenized microstructure. Films or discrete oxide particles form at melt surface and entrained within the melt during melt processing [13]. Due to the physical action of the shearing, these oxide films break up and disperse uniformly within the melt [18]. These oxide particles will enhance the nucleation of α-aluminum phase and refine the intermetallics [19]. This enhancement results in refined grains microstructure without any casting defect such as a segregation band.

#### 3.1.2. Extrusion Planks

The optical micrographs of TT-6111 (1), TT-6111 (2) and TT-6111 (3) planks extruded from homogenized or non-homogenized billets are shown in Figure 5. The micrographs show that the peripheral coarse grain was extensive for both planks extruded from homogenized DC-GR and MC-DC cast billets of TT-6111 (1), as shown in Figure 5a,b. Microstructural evaluation for planks extruded from homogenized billets of modified TT-61111 (2) showed a less peripheral coarse grain for DC-GR and MC-DC samples as shown in Figure 5c,d. It is very clear that the thickness of PCG zone was reduced and grain recrystallization was prevented across the planks with the increase of Mn, Cr, and Zr concentrations to 0.40, 0.08 and 0.05 wt.%, respectively. The co-addition of recrystallization inhibitors Mn, Zr, and Cr and eliminating the homogenization step had a favorable effect on the recrystallization of TT-6111 (3) alloy during the extrusion process with fibrous grains across the whole section of planks as shown in Figure 5e,f.

The peripheral coarse grain (PCG) is due to the formation of recrystallized grains layer in the upper and lower surface of the extruded part [20]. PCG is generally a surface deficiency observed in extruded aluminum alloys and that can result in poor surface quality with undesirable mechanical properties [21]. The alloy composition, presence of Mn, Zr, and Cr, billet microstructure and extrusion conditions can control the thickness of PCG across the extrusion section [22,23].

A fibrous structure enhances surface quality, forming characteristics, and corrosion resistance in addition to improving strength, toughness, and fatigue properties. Hence, the elimination of grain recrystallization across the whole section of extruded planks is important [22,24]. During processing of the billet, the recrystallization inhibitors form dispersoid particles that inhibit the nucleation and grain growth during further processing [25].

#### 3.1.3. Planks after Thermo-Mechanical Treatment 

The optical micrographs of annealed, cold rolled with intermediate heat treatment planks extruded from non-homogenized cast billets of TT-6111 (3) are shown in Figure 6 and Figure 7. The micrographs show fibrous structure, no recrystallization, and no PCG after cold rolling and intermediate heat treatment for all gauges.

### 3.2. Tensile Properties

#### 3.2.1. As-Cast Extrusion Plank

Figure 8 and Figure 9a show the mechanical properties of planks extruded from homogenized DC-GR and MC-DC cast billets of TT-6111 (2) for different orientations. The results show similar mechanical properties for both MC-DC and DC-GR casting in 0°, 90° and 45° orientations. The properties show that there is variation in total elongation between higher and lower values for DC-GR and MC-DC cast up to 19% and 10%, respectively for the different orientations tested. This anisotropy is well known for aluminum alloys sheets [26]. Okayasu et al. [27] found that the anisotropic microstructure that formed during sheet rolling control the mechanical properties of aluminum alloy sheets.

The mechanical properties of as-cast planks extruded from non-homogenized MC-DC cast billets of TT-6111 (3) in the transverse section were measured. Figure 9b shows the tensile results of as-cast homogenized and non-homogenized TT-6111 (2 and 3) from MC-DC casting. The results show the similarity of the tensile properties in both castings of TT-6111 (2) and TT-6111 (3). Eliminating the homogenization heat treatment step prior to extrusion has a benefit in terms of processing cost reduction.

#### 3.2.2. Planks after T4 and T8 Temper

With thermo-mechanical treatment (annealing, cold rolling and intermediate heat treatment) of planks extruded from non-homogenized MC-DC cast billets of TT-6111 (3), the results of tensile test in the T4P and the T8 temper for the three gauges (3 mm, 1.7 mm and 1 mm) are shown in Figure 10 and listed in Table 2. The yield strength and tensile strength increased with a decrease in ductility of T8 temper after paint bake simulation in comparison with the T4P temper.

Quainoo et al. [28] investigated the variation of yield strength as a function of aging time at 180 °C for various levels of cold work for AA6111 alloy. They found that as the aging time increases at each level of cold work, the yield strength values increases up to a peak value after which they decrease with further aging time. The significant increase in yield strength is attributed to the increase in dislocation density which piling up to form tangles and hence increasing the strength of the material. The time to reach peak strength for all levels of cold work occurred after about 10 h of aging. The yield strength after 2% stretch and a paint bake simulation at 180 °C for 30 min was 218 MPa [28]. However, for recycled TT-6111, the average yield strength after T8 temper was 258 MPa, which is a good achievement for our work.

The formability and final component quality of the alloy can be improved by optimizing the pre-aging and aging treatment conditions to lower the yield strength for stamping and then reducing of the natural aging kinetics of the stored product [29,30]. In our future work, we will optimize the thermo-mechanical treatment conditions for the recycled TT-6111 alloy to achieve the required specifications for automotive applications.

## 4. Conclusions

For the first time, the AA6111 aluminum automotive body-sheet alloy has been alloyed, casted, and processed to sheet successfully from 100% end-of-life recycled aluminum scrap.In comparison with the conventional DC and DC-GR castings of the TT-6111 alloy, the metallography of MC-DC cast showed refined equiaxed grains with no casting defects.Microstructural evaluation for planks extruded from non-homogenized of modified TT-6111 (3) showed eliminated peripheral coarse grain. The co-addition of recrystallization inhibitors (Mn, Zr, and Cr) with elimination of the homogenization step has a favorable effect on the recrystallization of AA6111 alloy during the extrusion process where a fibrous grain structure was retained across the whole section of the planks.The tensile properties of as-cast planks from non-homogenized billets were similar to those from homogenized billets. Eliminating the homogenization heat treatment step prior to extrusion has important ramifications in terms of processing cost reduction.The yield strength and tensile strength increased with a decrease in ductility of T8 temper after paint bake simulation in comparison with the T4P temper of TT-6111 alloy.

## Figures and Tables

**Figure 1 materials-13-05740-f001:**
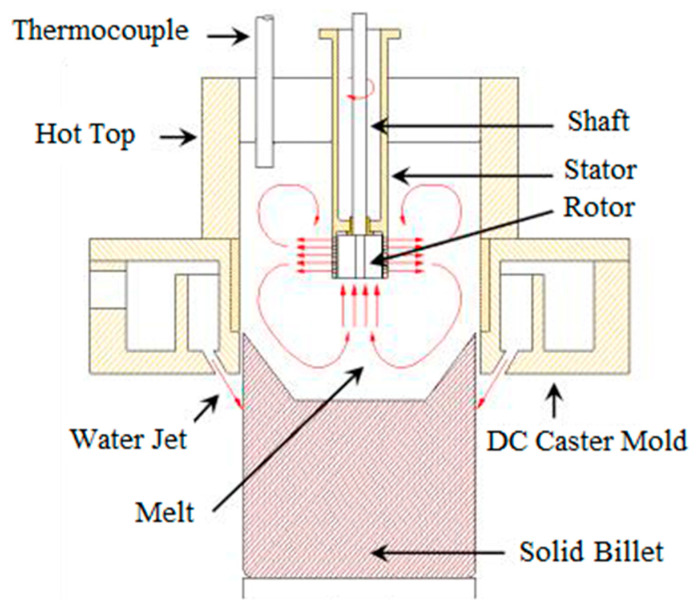
Schematic diagram of the melt conditioned direct chill casting (MC-DC) with the rotor-stator shearing device submerged in the sump of a conventional DC casting mold [1].

**Figure 2 materials-13-05740-f002:**
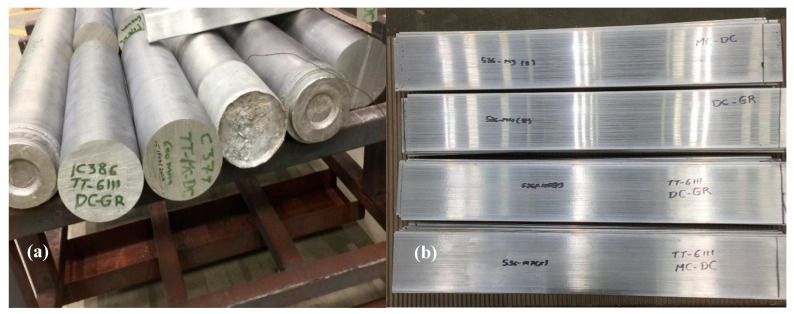
(**a**) DC, DC-GR and MC-DC Billets of TT-6111 alloy, and (**b**) Extruded flat bars produced from billets of TT-6111 alloy.

**Figure 3 materials-13-05740-f003:**
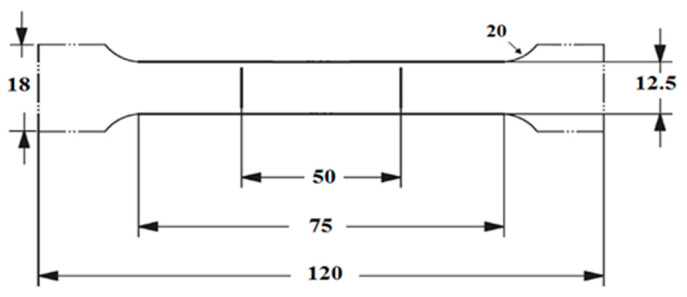
Dimensions of tensile specimen in mm according to BS EN ISO 6892-1:2016 standard.

**Figure 4 materials-13-05740-f004:**
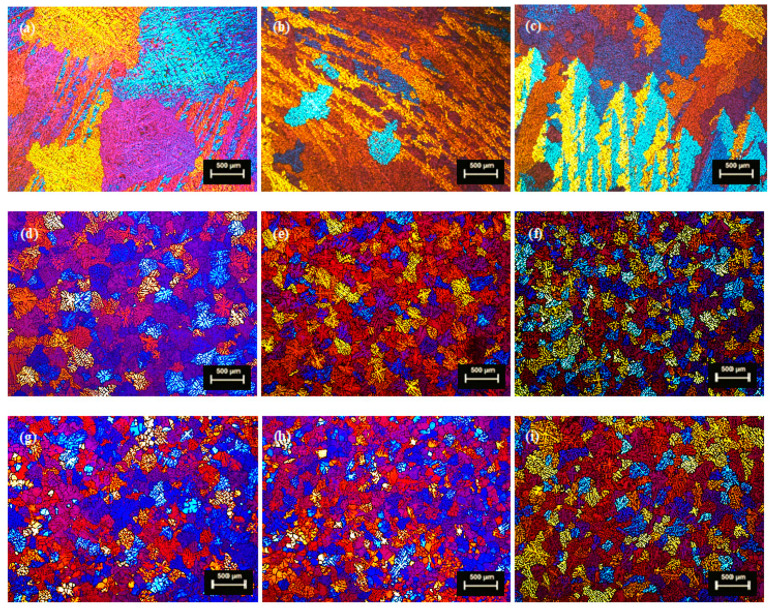
Microstructure of direct chill cast billets of Taint Tabor TT-6111 (3); (**a**,**d**,**g**) at the center, (**b**,**e**,**h**) half radius and (**c**,**f**,**i**) edge of billet. Where (**a**–**c**) DC billet, (**d**–**f**) DC-GR billet and (**g**–**i**) MC-DC billet.

**Figure 5 materials-13-05740-f005:**
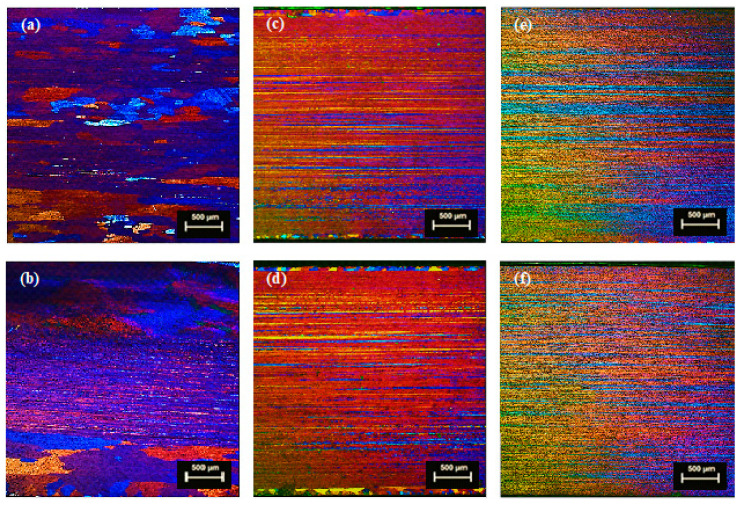
Optical micrographs of extruded planks in longitudinal direction; (**a**,**b**) formulated TT-6111 (1) extruded from homogenized billets; (**c**,**d**) formulated TT-6111 (2) with recrystallization inhibitors and extruded from homogenized billets; (**e**,**f**) formulated TT-6111 (3) with recrystallization inhibitors and extruded from non-homogenized billets. Where (**a**,**c**,**e**) DC-GR casting; and (**b**,**d**,**f**) MC-DC casting.

**Figure 6 materials-13-05740-f006:**
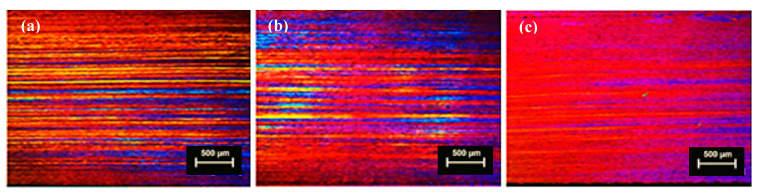
Micrographs of planks extruded from non-homogenized MC-DC of TT-6111 (3) billet with cold rolling and heat treatment in longitudinal direction; (**a**) annealing of as-cast planks at 360 °C for 2 h; (**b**) cold rolling of annealed planks to 3 mm; (**c**) annealing of 3 mm planks at 360 °C for 2 h.

**Figure 7 materials-13-05740-f007:**
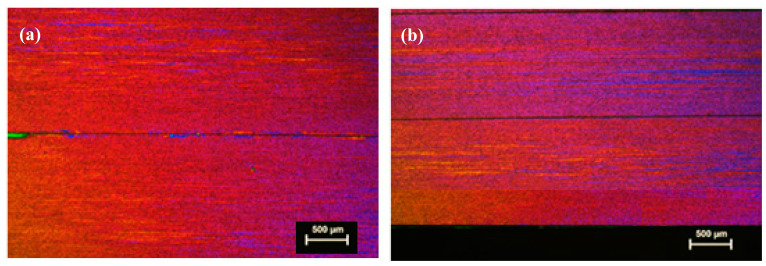
Micrographs planks extruded from non-homogenized MC-DC of TT-6111 (3) billet with cold rolling in longitudinal (**Top**) and transverse (**bottom**) directions; (**a**) cold rolling of annealed 3 mm planks to 1.7 mm and (**b**) cold rolling of 1.7 mm planks to 1.0 mm.

**Figure 8 materials-13-05740-f008:**
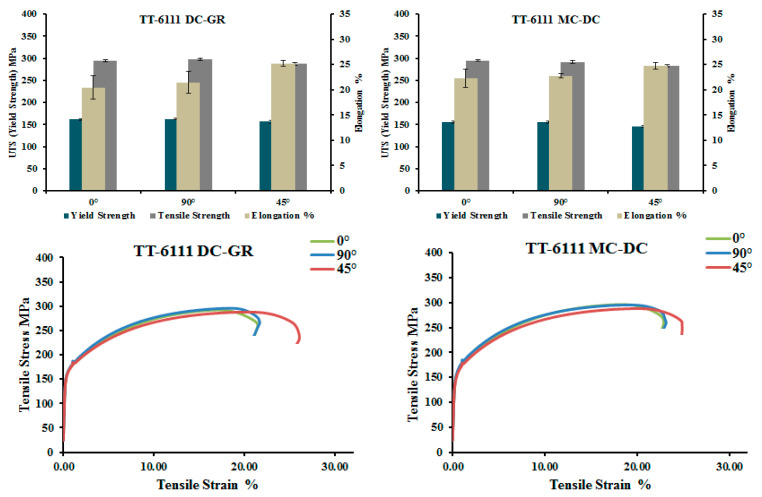
The tensile properties of as-cast TT-6111 (2) planks extruded from homogenized DC-GR and MC-DC cast billets in different orientations.

**Figure 9 materials-13-05740-f009:**
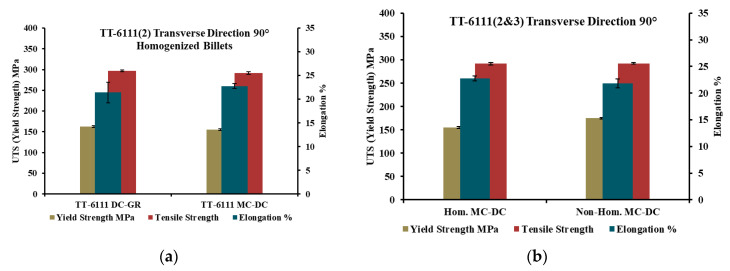
The mechanical properties of as-cast planks extruded from TT-6111 (2 and 3) billets in the transverse direction; (**a**) TT-6111 (2) planks extruded from homogenized DC-GR and MC-DC billets; (**b**) Comparison of the mechanical properties of TT-6111 (2) and TT-6111 (3) planks extruded from MC-DC casting billets.

**Figure 10 materials-13-05740-f010:**
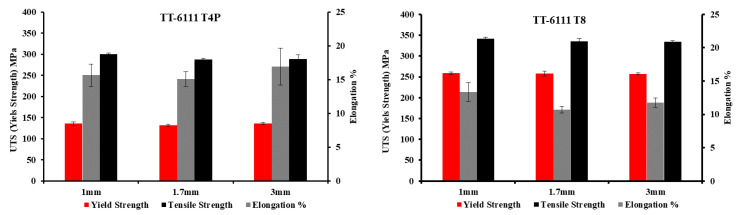
Tensile properties of non-homogenized TT-6111 planks, all variants, in T4P and T8 tempers.

**Table 1 materials-13-05740-t001:** Compositions of as-received TT ingots, formulated TT-6111 and target AA6111 alloys.

Wt. %	Norton TT (A052)	Formulated TT-6111 (1)	Target AA6111	Norton TT (F255)	Formulated TT-6111 (2)	Norton TT (F256)	Formulated TT-6111 (3)	Target Modified AA6111
Si	0.69	0.75	0.7–1.1	0.73	0.70	0.57	0.70	0.7–1.1
Fe	0.46	0.37	0.4 Max.	0.44	0.42	0.31	0.41	0.4 Max.
Cu	0.21	0.70	0.5–0.9	0.18	0.76	0.21	0.74	0.5–0.9
Mn	0.23	0.22	0.15–0.45	0.26	0.40	0.16	0.39	0.4
Mg	0.56	0.76	0.5–1.0	0.54	0.70	0.08	0.69	0.5–1.0
Cr	0.03	0.02	0.1 Max.	0.02	0.07	0.02	0.07	0.08
Ni	0.01	0.01	-	0.01	0.01	0.01	0.01	-
Zn	0.25	0.20	0.15 Max.	0.24	0.22	0.29	0.27	0.15 Max.
Ti	0.02	0.02	0.1 Max.	0.02	0.014	0.01	0.014	0.1 Max.
Pb	0.02	0.006	-	-	0.006	-	0.009	-
Sn	<0.01	0.001	-	-	0.001	-	0.001	-
Zr	-	0.002	-	0.002	0.047	-	0.05	0.05
Al	Bal.	Bal	Bal.	Bal.	Bal	Bal	Bal.	Bal.

**Table 2 materials-13-05740-t002:** Tensile properties of non-homogenized TT-6111 (3) planks after thermo-mechanical treatment in T4P and T8 temper for different gauge thickness in the transverse section for MC-DC castings.

Plank Thickness (mm)	TT-6111 Specimen Label	0.2% Offset Yield (MPa)		Tensile Strength (MPa)		Plastic Strain at Break (%)	
		T4P	T8	T4P	T8	T4P	T8
3	MC-DC	136	257	289	334	16.9	11.8
1.7	MC-DC	132	258	289	355	15.1	10.7
1	MC-DC	136	259	300	341	15.7	13.4

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
