# Peer review of "Direct Chill Casting and Extrusion of AA6111 Aluminum Alloy Formulated from Taint Tabor Scrap"

_materials, 2020, doi:10.3390/ma13245740_

Round 1

Reviewer 1 Report

Title: Direct Chill Casting and Extrusion of AA6111 Aluminum Alloy Formulated from Taint Tabor Scrap
By K.Al-Helal, et al

Recommendation: this paper is worth publication with minor revision.

Authors adopted MC-DC casting technology and conventional grain refinement by following extrusion and rolling to produce AA6111 from 100% Taint Tabor scrap Aluminum. At the meantime, they conducted
various evaluation measurements on metallography, microstructural, tensile properties and so on.

The manuscript is well structured. Their experimental findings, albeit, very fundamental not a breakthrough, support their conclusion. As far as a journal article is concerned, they are enough for publication. Weakness: they did not specify the industrial significance of their work. For instance, guidance on mass production, and real world applications, are not addressed as expected.

It might need to highlight practical applications on their obtained Aluminum alloy.

Author Response

Reviewer 1

The manuscript is well structured. Their experimental findings, albeit, very fundamental not a breakthrough, support their conclusion. As far as a journal article is concerned, they are enough for publication. Weakness: they did not specify the industrial significance of their work. For instance, guidance on mass production, and real world applications, are not addressed as expected.

It might need to highlight practical applications on their obtained Aluminum alloy.

Answer:

Dear Reviewer,

Thank you very much for reviewing my paper. I have mentioned in the introduction and conclusions about the application of AA6111 alloy in the automotive industry, it is highlighted in introduction (line 36) and conclusions (line 270).

(Done)

Kind Regards,

Reviewer 2 Report

The paper is interesting, the subject is worthy of investigation and the results have the potential to become a very interesting article. I appreciate authors' effort in performing the experiments. The research design is appropriate and the conclusions are supported by the results.

I have just a little observation: in Figure 5 the Tensile stress vs. Tensile strain graphs, the curves end a little bit unusual. It seems authors used in tensile testing a mechanical extensometer, especially since the strain rate was constant. If the answer is yes, please mention that in the manuscript at the end of the sub section 2.3 Thermomechanical Treatment and Tensile Testing.

Author Response

-Reviewer 2

I have just a little observation: in Figure 5 the Tensile stress vs. Tensile strain graphs, the curves end a little bit unusual. It seems authors used in tensile testing a mechanical extensometer, especially since the strain rate was constant. If the answer is yes, please mention that in the manuscript at the end of the sub section 2.3 Thermomechanical Treatment and Tensile Testing.

Answer:

Dear Reviewer,

Thank you very much for reviewing my paper.

The tensile machine is electromechanical and I have highlight it in the text (section 2.3, line 146).

Kind Regards

Reviewer 3 Report

The article is interesting and very up to date. Obtaining fully recycled aluminum is one of the many aspects necessary to reduce the problems of climate neutrality and reduce the consumption of natural resources. However, the presentation of results and analysis require a total redesign. 

propose that describe exactly the casting method (Figures indicated) Chapter 2 requires systematize when it is not clear how the conditions under which the sample was molded, a description of test methodology microstructure much better transfer to Chapter 3 Section results and the results are completely unreadable. 

I suggest a breakdown of Microstructure Analysis, Tensile Properties and Total Description (after casting, after extrusion, after heat treatment).

The presented photos of the microstructure require improvement. Please describe nazmienę which affects the structure after the subsequent processes and physical phenomena that occur (depending on the feedstock).

The mechanical properties presented in Figs. 5 and 6 should be summarized together and described with the obtained values. The analysis of Figs. 5 and 6 shows that the influence of the conditions under which the charge was obtained does not affect the obtained properties. Please comment.

If the authors write about the property anisotropy depending on the direction, what is the percentage difference of ?? Is there any grounds for such a statement, please give your opinion.

Why does the change in structure after heat treatment and mechanical properties result?

Are there any changes taking place, any chemical compounds being formed, if so what?

Conclusions 3, 4 and 5 have a weak foundation in the article itself.

The subject is very interesting, but please work on the readability of the presentation and the identification and connection of structure changes with mechanical properties. I would love to read the revised version of the article.

Author Response

-Reviewer 3

1-The article is interesting and very up to date. Obtaining fully recycled aluminum is one of the many aspects necessary to reduce the problems of climate neutrality and reduce the consumption of natural resources. However, the presentation of results and analysis require a total redesign.

Answer:

Dear Reviewer,

Thank you very much for reviewing our paper.

Regarding the design and structure of the manuscript, we used to write and publish our results in this style; the authors believe this style or design is more appropriate. May be for the second paper we will design the paper in another style.

2-propose that describe exactly the casting method (Figures indicated) Chapter 2 requires systematize when it is not clear how the conditions under which the sample was molded, a description of test methodology microstructure much better transfer to Chapter 3 Section results and the results are completely unreadable.

Answer:

-Regarding the casting method, we have explained it in details in section 2.1. We have many publications showing the details of melt conditioning direst chill casting and I have added one of these references to refer in section 2.1, line 101.

- For sample preparation and testing, we follow a standard procedure in analyzing Aluminum alloy samples.

3-I suggest a breakdown of Microstructure Analysis, Tensile Properties and Total Description (after casting, after extrusion, after heat treatment).

Answer:

Our and your breakdown can be followed, but we gave the priority to the process than the characterization.

4-The presented photos of the microstructure require improvement. Please describe nazmienę , which affects the structure after the subsequent processes and physical phenomena that occur (depending on the feedstock).

Answer:

-For microstructure micrographs quality, what we have presented are the best photos we can achieve.

- Sorry I dont know what do you mean by nazmiene?

5-The mechanical properties presented in Figs. 5 and 6 should be summarized together and described with the obtained values. The analysis of Figs. 5 and 6 shows that the influence of the conditions under which the charge was obtained does not affect the obtained properties. Please comment.

Answer:

- Regarding the results of mechanical properties, we mentioned about the similarity of DC-GR and MC-DC as shown in section 3.3.1, line 213.

6-If the authors write about the property anisotropy depending on the direction, what is the percentage difference of ?? Is there any grounds for such a statement, please give your opinion.

A- For anisotropy, we have clarify this and amended the text as highlighted in section 3.3.1, line 214.

7-Why does the change in structure after heat treatment and mechanical properties result?

Are there any changes taking place, any chemical compounds being formed, if so what?

Answer:

There are significant changes in mechanical properties after heat treatment as shown in Figure 9. For the microstructure, we will investigate it in more details in addition to the intermetallics form during heat treatment in near future.

8-Conclusions 3, 4 and 5 have a weak foundation in the article itself.

Answer:

Figures 4, 6 and 9 support our findings in conclusions 3, 4 and 5 and we discussed this in the manuscript. The results are novel and this is the first time to produced automotive alloy sheets from 100% scrap material with good quality. In addition to elimination of homogenization step which, will reduce the production cost and cut emission of CO2 .

9-The subject is very interesting, but please work on the readability of the presentation and the identification and connection of structure changes with mechanical properties. I would love to read the revised version of the article.

Answer:

Thank you for your comments. Our focus in this paper was on the successful of formulating, casting and extrusion of automotive alloy from 100% end-of-life recycled scrap. The second study will be on full characterization of the sheet microstructure and mechanical properties before and after heat treatment. in addition we will optimize the heat treatment conditions to achieve industrial requirement.

Kind Regards

Reviewer 4 Report

Comments and Suggestions for Authors

This study assessed the microstructure and tensile properties of AA6111 automotive body sheet formulated from Taint Tabor scrap. The end-of-life (TT) scrap was processed into billets in direct chill casting followed by extrusion process, cold rolling, heat treatment and paint bake simulation. Eliminating the homogenization heat treatment step prior to extrusion has important ramifications in terms of processing cost reduction. The manuscript has a good structure and evident results. However, it would be more complete to include data on the hardness of the different specimens since there is a clear relationship between hardness and microstructure.

I suggest the article to be accepted after minor revision.

Submission Date

29 October 2020

Date of this review

23 November 2020

Author Response

Reviewer 4

This study assessed the microstructure and tensile properties of AA6111 automotive body sheet formulated from Taint Tabor scrap. The end-of-life (TT) scrap was processed into billets in direct chill casting followed by extrusion process, cold rolling, heat treatment and paint bake simulation. Eliminating the homogenization heat treatment step prior to extrusion has important ramifications in terms of processing cost reduction. The manuscript has a good structure and evident results. However, it would be more complete to include data on the hardness of the different specimens since there is a clear relationship between hardness and microstructure.

I suggest the article to be accepted after minor revision.

Answer:

Dear Reviewer,

Thank you very much for reviewing my paper. I agree with you that hardness test is needed for full characterization. Due to lockdown and COVID-19 situation, we are not able to do this test now. We intend to do further investigation and optimize thermomechanical treatment conditions to achieve the required specifications for automotive applications. We will include hardness test within the next study.

Kind regards

Reviewer 5 Report

Understanding the need to recycle aluminum end-of-life scrap and the environmental concerns attached to it is of interest to the automotive industry.  The manuscript could be surely published in ‘Materials’ as most parts of the observations are reasonably correct and well discussed, as expected from this research group, the microstructure, mechanical properties, and tensile properties on AA6111 has been thoroughly carried out and analyzed. I recommend the paper be accepted for publication. I had the following comments:

  • It would be nice if the authors correlated this work with their previous work on AA6111 in the introduction and discussion.
  • Grammatical errors must be looked into carefully.

Author Response

Reviewer 5

It would be nice if the authors correlated this work with their previous work on AA6111 in the introduction and discussion.

Grammatical errors must be looked into carefully.

Answer:

Dear Reviewer,

Thank you very much for reviewing our paper.

Our previous work was on casting a recycled AA6111 alloy in high shear twin roll casting. The source recycled material and casting technique were different from present study.

Regarding the grammatical error, we have looked again and corrected any mistakes.

Kind regards

Round 2

Reviewer 3 Report

Thank You for answers, 

I understand that getting the described alloy is very important, but I will still stick to the position that the form of presentation could be much better. Moreover, adding a reference to line 101 in this way is inconsistent. As for the rest of the comments, I expected a little more insightful comments, maybe change the type of article to a research report ...

Author Response

Dear Reviewer,

I have redesigned the Results and Discussion section according to your suggestions. A schematic diagram has been added to show the melt conditioning direct chill process. In addition, I discussed the final results more deeply. Hope the paper now is more readable.

Many thanks for your useful comments.

Kind regards